# A Family of Cognitively Realistic Parsing Environments for Deep Reinforcement Learning

## Abstract

The hierarchical syntactic structure of natural language is a key feature of human cognition that enables us to recursively construct arbitrarily long sentences supporting communication of complex, relational information. In this work, we describe a framework in which learning cognitively-realistic left-corner parsers can be formalized as a Reinforcement Learning problem, and introduce a family of cognitively realistic chart-parsing environments to evaluate potential psycholinguistic implications of RL algorithms. We report how several baseline Q-learning and Actor Critic algorithms, both tabular and neural, perform on subsets of the Penn Treebank corpus. We observe a sharp increase in difficulty as parse trees get slightly more complex, indicating that hierarchical reinforcement learning might be required to solve this family of environments.

## 1 Introduction

We introduce a framework in which we can start exploring how reinforcement learning (RL; Sutton and Barto 2018) algorithms scale up against human cognitive performance, as captured by the syntactic parsing problem. Parsers grounded in contemporary generative linguistic theory involve rich, hierarchically structured representations and complex rule systems that pose significant challenges for RL algorithms. We begin with a simple example to illustrate the type of psycholinguistic task we modeled our environments on, namely self-paced reading tasks (Just and Carpenter, 1980; Just et al., 1982). In such tasks, the words are hidden and only one word is uncovered at a time with a spacebar press. The human reader decides when to press the spacebar to uncover the next word (which automatically hides the current word), hence the name of self-paced reading. Self-paced reading tasks mimic an essential aspect of naturally-occurring language comprehension with auditory stimuli: the signal is strictly linearly and strictly incrementally presented one word at a time. Just as in naturally-occurring verbal interactions, and unlike in normal reading situations, the linguistic signal cannot be 'rewound' to previous words – we cannot just look back and reread previous parts of the text – or 'fast-forwarded' to subsequent words – we cannot jump ahead to parts of the text that do not immediately follow the word currently being read.

We use a chart parser (Earley 1970; Tomita 1986; Scott 2008) with a cognitively-realistic eager left-corner parsing strategy (Resnik 1992; Hale 2014 a.o.) to provide the reward structure, thereby guiding the reinforcement learning process. Running this eager left-corner parser on a simple input sentence will shed light on its inner workings. Assume we have a simple grammar with three phrase structure rules (PSRs) S → NP VP, NP → Det N, and VP → V. Also, assume that we are reading the sentence *A boy sleeps* in a self-paced reading task. We start with a screen in which all words are covered with dashes: `- --- ------`. After the first space-bar press, the first word is revealed: `A --- ------`, and the parser recognizes its syntactic category Det (determiner) and takes a series of parsing steps that constructs the leftmost tree in Figure 1. We see here the left-corner nature of our parser: we trigger the PSR NP → Det N, which has Det as its left branch/corner, as soon as we recognize that the first word *A* is a determiner. This partial tree is only implicitly constructed in the chart parser: the chart does not store trees, but instead contains edges, which are left-corner based hypotheses about the possible syntactic structures we can associate with the linguistic input received so far. After another space-bar press, the noun is revealed (`- boy -----`), its syntactic category N is recognized and the richer partial tree shown in the middle of Figure 1 is constructed after a series of parsing steps. Finally, the verb is revealed after one more space-bar press: `- --- sleeps`, its syntactic category V is recog-

nized, and the final tree structure in Figure 1 is constructed.

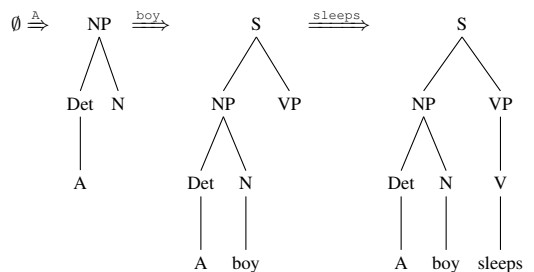

Figure 1: Partial trees built incrementally when reading the sentence *A boy sleeps* word by word

This parsing example shows that *proper action ordering is crucial to successfully completing the parsing process, which is like searching for a path through a maze:* (*i*) the position in the maze is the current parse state (the chart), (*ii*) the possible moves (up, left etc.) are the possible parsing steps, i.e., edges, we can add to the chart, and (*iii*) a path through the maze is given by the proper sequence of parsing steps / edges needed to successfully complete the parsing process.

Both humans and RL agents may get 'lost' in these parsing mazes, for example, when encountering a so-called 'garden path' example like *The horse raced past the barn fell* (Bever 1970 among many others). This sentence might seem ungrammatical, but it is in fact grammatical under the reduced relative clause interpretation that can be paraphrased as 'the horse that was raced past the barn fell.' To access that interpretation, one can compare the previous example with the sentence *The children taught by the Berlitz method passed the test* (Crain and Steedman 1985). This second example does not garden-path the reader because the most likely interpretation of *taught* in this sentence is the non-finite past-participle interpretation. This is in contrast to *raced* in the first example, which garden-paths the reader because its most likely interpretation is the incorrect, finite simple past one. This interpretation leads the reader down an incorrect garden path during incremental parsing.

In brief, parsing tasks can be viewed as executing a certain protocol, and RL is a family of methods to learn protocols. The paper makes two contributions. First, we introduce a new family of cognitively realistic parsing environments for RL that are ordered by the complexity of the parsing problems they pose. Second, we study the performance of 15 agents in the two easiest parsing environments

(height-4 and height-5 trees). The 15 agents fall into 3 classes: tabular, Deep Q Network (DQN) agents, and Actor-Critic (AC) agents; for the latter two, we experiment with LSTM (Hochreiter and Schmidhuber 1997), GRU (Cho et al. 2014), and Self-Attention networks (Vaswani et al. 2017).

## 2 Family of Parsing Environments

### 2.1 Parsing Actions

It is perhaps easiest to introduce our formalization of the parsing environment by examining an example episode perfectly played by a trained parsing agent, shown in Table 1. The sentence to be parsed is *He sighed*. The agent can take one of three types of actions. The first type of action is **scan**, which 'reads' the next word of the sentence. The words are 'read' one at a time, and the agent is not able to access the next word without a scan action. Furthermore, the final word at the end of the sentence is not explicitly marked as such: the agent learns that the end of the sentence has been reached only when it takes a scan action that fails, which receives a steep negative reward of $-2$. That is, the agent is actively encouraged to predict the end of the sentence.

The first two actions in Table 1 are scan actions, and we see that the agent correctly predicts the end of the sentence after the second word has been scanned and does not attempt to scan again. The results of these actions are two leaf edges which are added to the chart, and these edges are returned to the agent by the environment as (part of) the next state. Leaf edges consist of a span, which indicates what positions/parts of the input sentence are 'covered' by the leaf edge. The first scan adds a leaf edge spanning positions 0 to 1, that is, the first terminal, which is the word *He*. The second scan adds a leaf edge spanning positions 1 to 2, i.e., the second terminal, which is the word *sighed*. Actions that contribute to the final parse receive a small negative reward of $-0.25$, encouraging the agent to finish parsing as soon as possible.

The next five actions in Table 1 are **predict** actions. These actions target a complete edge in the chart (leaf edges are by definition complete; we discuss completion for other edges below), and identify a production in the grammar whose right-hand side starts with the terminal or non-terminal of that completed edge. That is, we identify productions whose left-corner is the targeted edge. For example, the first predict action (step 3 in Table 1), targets the leaf edge storing the word *He* and identi-

Table 1: A parsing episode perfectly played by a trained agent

| | action | | | state | reward |
|---|---|---|---|---|---|
| 1. | **scan** | | | [0:1] He | -0.25 |
| 2. | **scan** | | | [1:2] sighed | -0.25 |
| 3. | **predict**: | [0:1] He | PRP → He | [0:1] PRP → He ● | -0.25 |
| 4. | **predict**: | [0:1] PRP→ He ● | NP-SBJ → PRP | [0:1] NP-SBJ → PRP ● | -0.25 |
| 5. | **predict**: | [0:1] NP-SBJ → PRP ● | S → NP-SBJ VP | [0:1] S → NP-SBJ ● VP | -0.25 |
| 6. | **predict**: | [1:2] sighed | VBD → sighed | [1:2] VBD → sighed ● | -0.25 |
| 7. | **predict**: | [1:2] VBD → sighed ● | VP → VBD | [1:2] VP → VBD ● | -0.25 |
| 8. | **complete**: | [0:1] S → NP-SBJ ● VP | [1:2] VP → VBD ● | [0:2] S → NP-SBJ VP ● | 5.00 |

fies 'PRP → He' as a production whose left-corner is that word (PRP stands for personal pronoun). As a result of this action, we add a new edge to the environment chart, the one listed in step 3 under **state**. The span of this edge is 0:1, which is the same as the span of the edge targeted by the predict action (the edge covers only the first word of the sentence). The edge added in step 3 is not a leaf edge, as it builds syntactic structure on top of a previous edge. That syntactic structure is the unary branching node PRP, whose only daughter is the word *He*. This edge is a complete edge, indicated by the final dot ● after *He*: 'complete' means that the entire right-hand side of the production used to build the edge has been recognized, i.e., has been built out of complete edges.

The next predict action (in step 4) targets the edge '[0:1] PRP → He ●', which has just been added to the environment chart / state. This edge is a complete edge, and is also the left-corner of the production 'NP-SBJ → PRP;' NP-SBJ stands for subject noun phrase (NP). At step 4, we build more syntactic structure on top of the PRP non-terminal: this syntactic structure is encoded by the new edge '[0:1] NP-SBJ → PRP ●', which is a complete edge spanning position 0:1. That is, by end of the predict actions in steps 3 and 4, we have implicitly built the leftmost partial tree in Figure 2.

The predict action in step 5 targets the complete NP-SBJ edge we just added, which is identified as the left corner of the 'S → NP-SBJ VP' production. The resulting edge '[0:1] S → NP-SBJ ● VP ', which is part of the resulting **state** in step 5, is the first incomplete edge in this episode: the dot ● precedes the VP, indicating that only the NP-SBJ has been recognized. Note also that the span of this incomplete edge is still 0:1, as the span always indicates the part of the input sentence that has been completely recognized by the edge. At the end of step 5, we have implicitly build the second (partial) tree in Figure 2.

The predict actions in steps 6 and 7 build syntac-

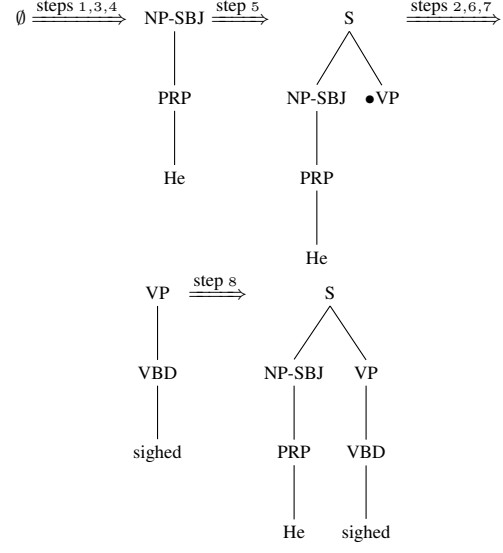

Figure 2: Partial trees implicitly built by the agent when playing the episode in Table 1

tic structure on top of the word *sighed* (VBD stands for verb in past form, and VP for verb phrase). The resulting tree is the third one in Figure 2.

The final action in the episode is a **complete** action. This is the third and final type of action that an agent can take. Complete actions target two edges in the chart (in the current environment state). The first edge is incomplete; specifically, this is the edge '[0:1] S → NP-SBJ ● VP' we built in step 5. The second edge has to be complete, and its left-hand side non-terminal has to be the very same as the leftmost incomplete non-terminal on the right-hand side of the incomplete edge. This final action completes the parse, building the complete final tree in Figure 2. The reward in this step is a substantial positive reward of 5.

In sum, agents can take one of three types of actions: scan, predict and complete. Predict actions target an edge in the current chart that can be the left corner of a grammar production. Complete actions target two edges in the current chart, an incomplete edge and a complete edge; the complete

edge can be used to bring the incomplete edge one step closer to completion. States returned by the environment consist of the current chart/list of edges, which encode the partially built syntactic structures in a very compact manner. Charts efficiently encode multiple complete parses if the sentence to be parsed is syntactically ambiguous, as in the typical prepositional phrase (PP) attachment example *I saw the astronomer with a telescope*. The PP *with a telescope* can be attached to the noun *astronomer* (the astronomer has a telescope), or to the verb *saw* (the seeing was done by means of a telescope).

## 2.2 Environment Setup

For every episode, the agent is tasked with parsing a single sentence. In all our experiments, the sentences come from the parsed Brown Corpus part of the Penn Treebank-3 Corpus (Marcus et al., 1999). The specific set of parse trees we load into our parsing environment determine the difficulty of the parsing tasks an RL agent will face. This enables us to create a wide variety of parsing environments that can be finely tuned in terms of difficulty. The level of difficulty is determined by the kind of tree structures we allow in the input set of trees. One way to decrease the level of difficulty is to restrict the kind of syntactic structures we allow; for example, removing trees that contain adjuncts decreases the level of difficulty. Another way is to limit the height of the trees, where tree height is defined as the length of the longest path in the tree starting at the root. For example, our experiments were run first with height-4 trees only, which is the smallest height with a reasonable number of trees (more than 150), after which we investigated height-5 trees. We have created a variety of tree sets along these lines. Height-6 trees, for example, even without adjunct structures, already raised the level of difficulty to a point that exceeded our computational resources (3 separate GPUs, the best of which was a Titan RTX, not always accessible).

Once a subset of trees is identified (by calibrating it for tree height, types of syntactic structures, and sometimes maximum sentence length) and loaded into the parsing environment, it is split into train, validation and test sets according to percentages provided by the user. The train-validation-test split is determined by a random seed that can be set by the user for reproducibility. Given our limited computational resources and the pilot benchmarking nature of this work, we couldn't do a systematic hyperparameter search, so we report results based only on train-test 90%-10% splits, without a separate validation set for hyperparameter tuning. We set the hyperparameters to values that seemed reasonable (often defaults); see next section.

After the set of trees is loaded, and before the train-test split is determined, the environment creates a context-free grammar (CFG) based on the productions implicit in all the loaded trees. The environment uses this CFG to generate the reward structure for any given episode. Every training episode consists of the agent learning to parse one sentence from the train set. What we do is create a 'maze' based on each individual sentence, and train agents on these 'mazes.'

For every train sentence, the environment uses the CFG to left-corner chart-parse the sentence and identify all its possible parses, as well as all the complete and incomplete parse edges that contribute to these parses. These parse edges enable us to generate the reward structure for every episode: if the agent takes an action resulting in an edge that contributes to one of the possible parses, the agent receives a small negative penalty, which was $-0.25$ in our experiments, but this, and all other rewards, can be set by the user. If the agent selects a parsing action that is licensed by the current state (which consists of the current chart and whether there are still words that need to be scanned), but does not contribute to one of the possible parses, the agent receives a larger negative penalty of $-0.75$. Actions are licensed if they create a valid edge that can be added to the current chart ('valid' based on the current chart and the background CFG). If the agent selects a parsing action that is not licensed by the current state, for example, it tries to add an edge that was already added to the chart, or tries to scan a word when there are no more words to be scanned, the agent receives a steep negative penalty of $-2$. Finally, when the agent selects an action that adds the final edge needed to complete a full parse of the sentence, it receives a positive reward of $5$ and the episode terminates. An optimal policy takes the minimum number of actions necessary to construct a complete parse of the input sentence.

While agents attempt to build alternative parse structures from those validated by the PTB-based CFG, they are negatively penalized by the environment for all the edges that don't contribute to any valid parse. Due to charts being able to compactly represent multiple parse trees, it's always possible

for an agent to eventually arrive at the correct parse by exhaustively executing all parse actions offered by the environment. However, this would result in a significantly lower total reward than taking the shortest, most cognitively realistic path to the correct parse (as seen in Sec. 4 Results).

Agents are trained to take the minimum steps possible because this heuristic is part of what the human parser does (Hale 2011 a.o. and references therein). It is precisely this minimum-cost feature of the human parser that leads it down garden paths. Hale (2011) provides suggestive evidence that a distance metric (i.e. an estimate of steps necessary to complete a parse) inferred based on PTB counts can be used to guide parsing in a way that captures a variety of garden-path phenomena. The present work is a first step towards using RL methods to learn this metric from experience (building on Hale 2014), and in the process, hopefully provide evidence for the cognitive realism of this minimal-effort / minimal-cost hypothesis.

As already indicated, the environment maintains a chart, which starts out empty for each sentence/episode. This chart, together with a Boolean indicating whether all words have been scanned or not, forms the state that the environment returns to the agent after every action. In our experiments, we decreased the difficulty of learning and provided the agent with the list of all possible predict and complete actions licensed by the environment in any given state. This simplifies the learning problem, as it effectively reduces action generation to action selection from a provided set of choices. The simplification can be easily removed, which would force the agent to *generate* actions.

## 3 Experiments

### 3.1 Agent Architectures

We study the performance of 15 RL algorithms / agents in the height-4 and height-5 parsing environments: (*i*) a tabular $Q$ learning agent (Watkins 1989; Watkins and Dayan 1992), (*ii*) 7 DQN agents (Mnih and al 2015, Sutton and Barto 2018, Ch. 11 and references therein), and (*iii*) 7 Actor-Critic (AC) agents (Sutton and Barto 2018, Ch. 13 and references therein). The 7 DQN agents differ with respect to their architecture, and so do the AC agents. Six of them are recurrent: Elman/standard RNN (Elman 1990), GRU (Cho et al. 2014), LSTM (Hochreiter and Schmidhuber 1997), and bidirectional versions of these three (Schuster and Paliwal

1997; Graves and Schmidhuber 2005). We also study a self-attention (Vaswani et al. 2017) agent. The learning rate $\alpha$ was always set to $10^{-3}$ and the discount factor $\gamma$ to 0.9.

The tabular $Q$ agent represents the $Q$ function as a look-up table that stores the estimated values of all possible state-action pairs. The state $s_t$ at time step $t$ consist of all the edges in the current chart, plus a Boolean indicating whether all the words have been scanned, i.e., whether we reached the end of the sentence. The action $a_t$ selected by the agent in state $s_t$ can be a scan, predict or complete action, as discussed in the previous section. Before learning begins, all the entries in the $Q$ table are set to an arbitrary value (0), and they are updated in an entry-wise fashion at each time step $t$: the value of the pair $(s_t, a_t)$ is updated based on the reward signal $r_{t+1}$ and the new state $s_{t+1}$ that the agent receives from the environment after taking action $a_t$. The new state $s_{t+1}$ updates the chart in the previous state $s_t$ with the new edge (if any) added by action $a_t$.

DQN agents approximate the $Q$ function with an artificial neural network (ANN). Their basic architecture is provided in Fig. 3. The edges in the current chart are numericalized (we return to this in a moment) and the resulting tensors are the input to a recurrent or self-attention ANN, the output of which is a chart tensor that 'summarizes' the current chart. We have a variety of choices for how to compute the chart tensor, but we only explore the simplest choices here: for RNNs, we take the chart tensor to be the final hidden state (or the two final hidden states for bidirectional RNNs); for self-attention, we mean-pool the attention outputs. This chart tensor is then concatenated with the numericalized action we're evaluating, and the resulting tensor is the input to a multilayer perceptron (MLP) with a single hidden layer and a ReLU nonlinearity. The output of the MLP is the predicted $Q$ value for the current state (chart) and action. We use an $\epsilon$-greedy policy, with $\epsilon$ annealed from a starting value of 1 to a minimum value of 0.01. All DQN agents were trained using one-step semi-gradient TD (a.k.a. semi-gradient TD(0); Sutton and Barto 2018, Chapters 9-11), with a squared TD-error loss and the Adam optimizer (Kingma and Ba 2015).

The AC agents use the same architecture as the DQN agents for their policy-approximation component: the single-value output is now the estimated logit for the action we're evaluating. The

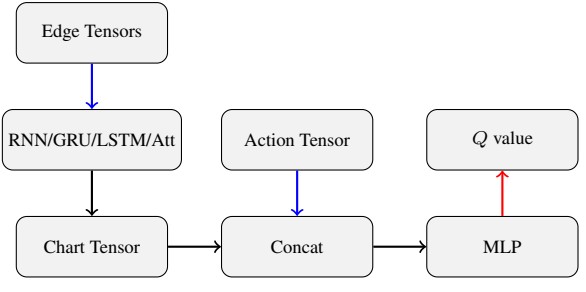

Figure 3: The basic architecture of the DQN agents

logit is soft-maxed together with the logits for all the other actions valid in the current state to yield a probability distribution over these actions. To this main policy-approximating branch, we add a separate state-value estimation MLP head with a single hidden layer that takes the chart tensor as input and outputs the estimated value for the current state (chart). The state-value head is trained using a squared TD-error loss (we backpropagate these gradients only through the state-value MLP head). The main policy-network branch is updated using a one-step version of REINFORCE (Williams 1992; see also Sutton and Barto 2018, Ch. 13 and references therein). All the recurrent and self-attention networks had a state/query/key/value size of 256, which was also the hidden-layer size of all MLPs.

There are many possible choices for edge-to-tensor numericalization, including trainable embeddings. In our experiments, we used a simple, deterministic algorithm. As discussed in the previous section, an edge is basically a CFG production with three associated integers: two integers are used to indicate the span of the edge (the part of the sentential input that is 'covered' by the edge), and the third one is used to indicate the dot position, i.e., how much of the right-hand side of the production has been completed. To numericalize an edge, we need to decide how to numericalize the non-terminals and terminals in the production part, and these three integers. For the integers, we used a one-hot encoding with a dimensionality equal to the maximum length of the right-hand side of a production in our CFG. For the non-terminals, we also used a one-hot encoding with the dimensionality provided by the number of non-terminals in our CFG. For terminals, i.e. words, we used GloVe embeddings (Pennington et al. 2014) and reduced their dimensionality to the dimensionality of the non-terminal one-hot encodings via a principal component analysis (PCA) model. For out-of-vocabulary (UNK) words without GloVe

embeddings, we used the mean GloVE embedding. Actions were numericalized using the same algorithm. For complete actions, we concatenated the numericalizations of the two edges in the complete pair. For predict actions, we numericalized the edge and then numericalized the production as if it was an edge, with null tensors for the three spurious integers. For scan actions, we concatenated two null edge tensors of the appropriate dimensionality.

## 3.2 Environment Specifics

We select the two simplest parsing environments we can create based on the Brown part of the Penn Treebank corpus: an environment based on a subset of the trees of height 4, and an environment based on a subset of the trees of height 5. To make the difficulty as low as possible for the RL agents, the trees are selected so that they always begin with a subject NP. The height-4 environment had 178 trees, with a random $90\% - 10\%$ train-test split that varied across 4 experimental runs; all the results reported in this section average across these runs, hence, across multiple random train-test splits. All splits had 160 train sentences and 18 test sentences. We did not have a separate validation set: given our limited computational resources, we didn't do any hyperparameter tuning (except for a handful of very limited comparisons). The height-5 environment had a 776 trees, and the $90\% - 10\%$ split had 698 train sentences and 78 test sentences.

Although these two environments are the simplest in terms of difficulty, the jump in difficulty from height 4 to height 5 is significant (and it only gets more substantial when moving to height-6 trees etc.). The maximum sentence length for height-4 trees was 7, while for height-5 trees, it was 12. The CFG induced by the height-4 trees had only 269 productions with a maximum right-hand side length of 4, while the CFG induced by the height-5 trees had 1761 productions with a maximum right-hand side length of 7. Because of this, the average number of parses for a sentence from the height-4 corpus (according to the induced CFG) was 1.09, while the average number of parses for a sentence from the height-5 corpus was 18.14. The number of parses increases very quickly as tree height goes up. For example, height-6 based CFGs associate more than 35 million parses with relatively short sentences like *The pale blob of the woman disappeared*. Yet another way to see the jump in difficulty from height 4 to height 5 is to

compare the average number of valid predict and complete actions per step: for height 4, there are on average 4.5 predict actions and 1.11 complete actions per step, while for height 5, there are on average 97.39 predict actions and 1.82 complete actions per step.

The performance of the agents in the height-4 and height-5 parsing environments are provided in Tables 2 and 3. On height-4, the agents were trained for 15,000 episodes, while on height-5, they were trained for 5,000 episodes. There are fewer episodes for height-5 because the episodes are much longer than the height-4 ones; 5,000 height-5 episodes are roughly equivalent to 15,000 height-4 episodes in that the $\epsilon$-annealing schedule has a similar profile relative to a full training run. The results in these tables (both means and standard errors) are averaged over 4 independent runs. The height-5 results for some of the AC agents were computed on less than 4 runs because of their substantially higher computational-resource demands. The number of steps was also limited for some of the AC agents on height 5. Based on a very small set of comparisons, step-limiting did not hurt the AC agents' training.

### 3.3 Evaluation

To better evaluate the agents, we estimated a floor and a ceiling for their performance in these environments. The floor is provided by agents randomly choosing an action in any state from the set of actions that are valid in that state. For height 4, the random agent achieves an average total reward per episode of $-4.87$, obtained in $18.14$ steps per episode (on average). For height 5, the random agent achieves an average total reward of $-96$ in $156.21$ steps per episode.

We estimate the performance ceiling for any given sentence by looking at all the possible parses of the sentence based on the environment CFG, and all the complete and incomplete edges that contribute to any of these parses. With the edges and parses in hand, we can compute the average reward per parse by multiplying the number of edges by $-0.25$ (which is the cost of any parsing action that contributes to a successful parse), adding 5 for all final edges (this is the final reward for completing the parse), and dividing by the total number of possible parses. We can compute the average minimum number of steps in the same way (we divide the number of edges by the number of

parses). These estimates are fairly accurate for low-ambiguity sentences like the ones in the height-4 environment, but they tend to be overly optimistic for higher-ambiguity sentences like the ones in the height-5 environment. To see this, take the typical PP-attachment ambiguity example *I saw the astronomer with a telescope*. The average number of minimum steps to a successful parse out of 2 possible parses is probably higher than the number of edges contributing to either of those parses divided by 2: most of the edges have to be added for either one of the parses, which only differ with respect to a small number of edges. This being said, the estimated average maximum reward for the height-4 environment is $2.99$, and the average minimum number of steps is $10.05$, which are very likely close to the true values because of the low ambiguity of height-4 sentences. For height 5, the estimated average maximum reward is $4.13$, and the estimated minimum number of steps is $5.47$. These height-5 estimates are likely pretty far from the true values, for which the average number of minimum steps seems closer to 30, which puts the average max reward for height 5 closer to $-2$.

## 4 Results

With these performance ranges in mind, we can turn to a discussion of the results in Tables 2 and 3. We see that overall, DQN agents outperform AC agents, with the performance of tabular agents being the poorest. The tabular agent performs about as well as the random baselines in both the height-4 and the height-5 environments. Since the tabular agent effectively memorizes the training data, it only very slightly generalizes from train to test in height 4 (many of the trees have similar subparts, so a small amount of generalization is possible), but completely fails to generalize in the more difficult height-5 environment.

On height 4, the DQN agent with an Elman (simple) RNN is the best on the test sentences; see the two boldfaced numbers in the left half of Table 2. This is likely because the other agents end up overfitting the training data. The performance of the DQN RNN agent is very close to ceiling performance, i.e., to the estimated max reward and min steps per episode for height 4, indicating that, for all intents and purposes, we have solved the height-4 environment. The DQN GRU agent is also a solid performer in height 4, trailing behind DQN RNN only slightly.

Table 2: Q-learning agents: mean total rewards / steps (and standard errors) on train / test

| Agent | Height 4 | | | | Height 5 | | | |
|---|---|---|---|---|---|---|---|---|
| | Reward | | Steps | | Reward | | Steps | |
| | Train | Test | Train | Test | Train | Test | Train | Test |
| Tabular Q | -4.73 (0.05) | -4.18 (1.06) | 19.15 (0.08) | 18.17 (1.69) | -124.57 (1.1) | -127.59 (8.98) | 200.25 (1.62) | 204.75 (13.27) |
| DQN RNN | 0.13 (0.04) | **2.56** (0.32) | 13.51 (0.06) | **10.26** (0.57) | -28.84 (0.63) | -22.71 (4.24) | 63.48 (0.92) | 55.08 (6.27) |
| DQN GRU | 0.93 (0.03) | 2.33 (0.38) | 12.59 (0.05) | 10.86 (0.73) | -24.48 (0.56) | -16.36 (2.87) | 56.37 (0.83) | 43.84 (4.44) |
| DQN LSTM | 0.96 (0.03) | 1.65 (0.78) | 12.53 (0.05) | 11.93 (1.3) | -27.07 (0.6) | -19.75 (3.62) | 60.81 (0.88) | 50.55 (5.48) |
| DQN Bi-RNN | 0.11 (0.04) | 1.87 (0.53) | 13.55 (0.06) | 11.46 (0.92) | -29.95 (0.64) | -22.84 (4.83) | 64.88 (0.94) | 53.98 (6.97) |
| DQN Bi-GRU | 0.96 (0.03) | 1.84 (0.65) | 12.52 (0.05) | 11.67 (1.1) | -20.48 (0.3) | **-15.09** (2.91) | 48.62 (0.42) | **41.86** (4.37) |
| DQN Bi-LSTM | 0.91 (0.03) | 2.17 (0.57) | 12.62 (0.05) | 11.08 (1.02) | -22.78 (0.3) | -21.52 (4.52) | 52.8 (0.41) | 53.02 (6.75) |
| DQN Self-Att | 0.75 (0.03) | 1.94 (0.45) | 12.83 (0.05) | 11.71 (0.87) | -25.44 (0.58) | -20.43 (3.98) | 58.92 (0.86) | 51.44 (5.92) |

Table 3: Actor-Critic agents: mean total rewards / steps (and standard errors) on train / test

| Agent | Height 4 | | | | Height 5 | | | |
|---|---|---|---|---|---|---|---|---|
| | Reward | | Steps | | Reward | | Steps | |
| | Train | Test | Train | Test | Train | Test | Train | Test |
| AC RNN | -0.61 (0.04) | 0.59 (1.22) | 14.38 (0.06) | 12.78 (1.81) | -44.1 (0.26) | -60.64 (5.33) | 79.24 (0.34) | 109 (7.77) |
| AC GRU | 1.81 (0.03) | 1.39 (0.73) | 11.63 (0.04) | 12.19 (1.14) | -47.38 (0.74) | -38.92 (4.1) | 90.39 (1.07) | 78.45 (6.13) |
| AC LSTM | 1.74 (0.03) | 0.61 (1.15) | 11.67 (0.04) | 13.44 (1.78) | -36.72 (0.28) | -45.4 (4.75) | 71.12 (0.38) | 86.44 (7.01) |
| AC Bi-RNN | -0.59 (0.04) | 0.3 (0.86) | 14.44 (0.07) | 13.46 (1.32) | -70.29 (0.79) | -70.96 (6.51) | 121.69 (1.16) | 123.22 (9.68) |
| AC Bi-GRU | 1.83 (0.03) | 1.37 (0.89) | 11.63 (0.04) | 11.9 (1.4) | -31.19 (0.28) | -34.66 (5.02) | 64.54 (0.39) | 71.37 (7.3) |
| AC Bi-LSTM | 1.34 (0.03) | 1.3 (0.73) | 12.15 (0.05) | 12.26 (1.17) | -35.73 (0.27) | -47.69 (6.14) | 70.59 (0.37) | 91.44 (8.96) |
| AC Self-Att | -2.08 (0.04) | -0.64 (0.72) | 15.87 (0.07) | 13.64 (1.2) | N/A | N/A | N/A | N/A |

However, the extra capacity in the more complex agents is helpful in the height-5 environment, where the bi-directional GRU performs the best; see the two boldfaced numbers in the right half of Table 2. Once again, the DQN GRU is a solid performer, trailing only slightly behind its bidirectional cousin on height-5. However, even the performance of the best agent (DQN Bi-GRU) is not at ceiling on height 5. As mentioned before, we have not done a hyperparameter search because of our limited computational resources, but the sub-ceiling performance of the best agent does not seem to be due to model capacity (only): we compared a DQN RNN agent with double the hidden state in the recurrent component (512) and double the hidden-layer size in the MLP component (512 again); its performance on height 5 was not distinguishable from the DQN RNN agent in Table 2.

We see that these cognitively realistic parsing environments provide a substantial challenge for current RL algorithms – recall that the height-4 and height-5 environments provide the simplest levels of difficulty, which very abruptly escalates for height 6 and above. A minimal increase in difficulty would be to use all the height-4 and height-5 trees, not only the ones starting with an NP subject. The AC agents performed more poorly than the DQN agents, which might be due to the fact that DQN is probably more sample efficient than AC. This is particularly interesting given that AC architectures have been argued to be cognitively realistic

(Botvinick et al. 2009). The self-attention AC agent in particular performed surprisingly poorly. Self-attention DQN is a pretty solid, middle-of-the-pack performer on both height 4 and height 5. The training of the self-attention AC agent, however, was unstable, which resulted in poor performance on height 4 and a complete lack of convergence in the more difficult height-5 environment. We are still diagnosing this issue.

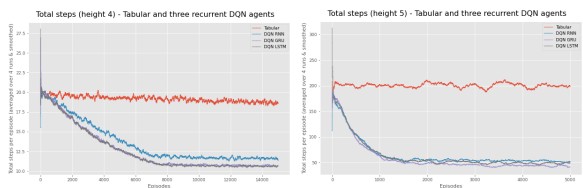

Figure 4: Steps per episode for tabular and three DQN agents training on height-4 (left) and height-5 (right)

The plots in Fig. 4 show the number of steps per episode (lower is better) for four agents, the tabular one and the three simplest recurrent DQNs. We see in both of them that the tabular agents seems to learn, but very slowly, while the DQN agents learn much faster. However, the DQN agents top out at a sub-ceiling level of performance on height 5, indicating that this environment remains unsolved. We leave an extensive hyperparameter search for a future occasion.

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
