# OpenReview forum: "A Family of Cognitively Realistic Parsing Environments for Deep Reinforcement Learning"
_aclweb.org/ACL/2022/Workshop/CMCL — Submitted to CMCL 2022_

### Official Review · Reviewer_xqgf · 2022-03-20
**This paper presents an approach to left-corner parsing of constituency structure (as CFGs) based on Reinforcement learning. The authors explore various RL environments, and evaluate them with respect to several baselines. However, the paper is lacking in motivation, and there is no clear placement of the authors' question/motivation/techniques with respect to the broader cognitive modeling literature.**

**Rating:** 4
**Confidence:** 4

**Review:**

To the Editors: I feel compelled to state that I had already reviewed this paper for a previous conference several months ago, and none of the issues found there have been addressed. Because of this, I can't confidently believe the authors' would  take the reviews into account in a potential camera ready. The rest of my review is the same as my previous one, since nothing in the paper has been changed.

## Big Picture Comments

From one side, the paper is generally well-written and the technical results seem sound (it is not possible to replicate the results, since the authors do not share their code, but the technical set-up appears correct to me, independently of the specific numerical scores).
The contribution is **potentially** interesting, but in ways that are made hard to evaluate given the paper.
Unfortunately, the main issue with this paper  its lack in clarity with respect to aims and overall conceptual contribution. In short: it is unclear what we are actually learning from the paper, and who the authors' think should be interested in their results (psycholinguists?computational linguists interested in symbolic models? neural network researchers? all of the above?). This is partially an artifact of significant presentation issues. I hope my comments below might help the authors' clarify the goals of the paper and its contribution to the broader literature.

## Specific Comments

One issue in presentation is, in my opinion, the focus of the first half of the paper given the audience. The authors spend almost half of the paper motivating constituency parsing, and walking us through an example of the shift/predict/scan operations. This to me seems quite trivial, and if the purpose is to exemplify the rewards, that's anyway made clearer in the later section.
However, what is missing is the motivation behind why applying a RL approach to this problem is valuable, and a walkthrough of RL fundamental intuitions, which seems to me way more needed given the scope of the paper than the scan/predict example.

I know it seems I am being picky about form, but the reason is that this lack of motivations makes the rest of the paper relatively weak.
It seems like the whole contribution is to substitute the classical control structure of these kind of parsing algorithms with a RL agent. Which, fine, but what is the advantage?
The authors seem to be implying that the contribution is showing that the economy heuristic guiding the agents is learned from experience, but that just doesn't seem to be true in any relevant sense. The fact that shortest parsing paths are preferred is baked into the reward structure, both at the local choice level and in terms of total reward, which explicitly penalized longer paths. So ok, this is less explicit than an oracle considering number of actions explicitly, but I would not say it "derives" from experience.

In turn, this leads the experimental section to land quite flatly. The authors claim implementing these kind of algorithms is challenging for RL. But so what is the contribution meant to be? In the introduction it seemed that the authors were arguing for advantages coming from applying RL to this parsing "problem". But then towards the end it seems that the contribution is in highlighting the limits of RL?
I am not saying these are not interesting results per se, they could be, but the authors never state why they should.

In short,  I think the conceptual foundation is missing from the paper, and thus the reader is left wondering. In this sense, the paper could benefit from a conclusion/summary section moving back from the particular results of the last section to the broader contribution. Clarifying what the authors think the aim of their approach is.


Some minor notes:

- The authors put a lot of emphasis on the parsers being "cognitively plausible", and I think they mean two things: (a) the eargerness of the left corner strategy (cue Resnik) and (b) the shortest path (parsing step counting) heuristic. Is that correct? I have no problems with either, in principle. But I don't think the prominence of either in the actual suggested contribution justifies having "cognitively plausible" in the title. The referenced literature seems to be lacking in this sense, missing a lot of work on connecting symbolic parsing algorithms to sentence processing behavior (boston2008parsing,chen2021quantifying,stanojevic2021modeling) Also note that there is a variety of ways (depending on what kind of cognitive, representational commitments one entails) in which these kind of minimal-steps ideas can be understood.

- In the abstract and through, the authors mention evaluating the"psycholinguistic implications" of RL algorithms, but there is barely a mention of that towards the end (that is left uninterpretable given the content of the paper). It is also left unclear how actually plausible these algorithms are, since the tree depth and things like adjunction seem to matter in ways that clearly do not for the human parser (sturt2005processing). In fact, we are even left wondering what the "cognitively plausible" mentioned in the title refers to.

## Some References

@article{boston2008parsing,
title={Parsing costs as predictors of reading difficulty: An evaluation using the Potsdam Sentence Corpus},
author={Boston, Marisa Ferrara and Hale, John and Kliegl, Reinhold and Patil, Umesh and Vasishth, Shravan},
journal={Journal of Eye Movement Research},
volume={2},
number={1},
year={2008}
}

@article{chen2021quantifying,
title={Quantifying Structural and Non-structural Expectations in Relative Clause Processing},
author={Chen, Zhong and Hale, John T},
journal={Cognitive Science},
volume={45},
number={1},
pages={e12927},
year={2021},
publisher={Wiley Online Library}
}

@inproceedings{kobele2013memory,
title={Memory resource allocation in top-down minimalist parsing},
author={Kobele, Gregory M and Gerth, Sabrina and Hale, John},
booktitle={Formal grammar},
pages={32--51},
year={2013},
organization={Springer}
}


@inproceedings{stanojevic2021modeling,
title={Modeling incremental language comprehension in the brain with Combinatory Categorial Grammar},
author={Stanojevi{\'c}, Milo{\v{s}} and Bhattasali, Shohini and Dunagan, Donald and Campanelli, Luca and Steedman, Mark and Brennan, Jonathan and Hale, John},
booktitle={Proceedings of the Workshop on Cognitive Modeling and Computational Linguistics},
pages={23--38},
year={2021}
}

@article{sturt2005processing,
title={Processing coordinated structures: Incrementality and connectedness},
author={Sturt, Patrick and Lombardo, Vincenzo},
journal={Cognitive Science},
volume={29},
number={2},
pages={291--305},
year={2005},
publisher={Wiley Online Library}
}

---

### Official Review · Reviewer_DZ3M · 2022-03-26
**Unclear motivation**

**Rating:** 3
**Confidence:** 4

**Review:**

The paper describes a method for training a left-corner parser through reinforcement learning using word vectors from a variety of off-the-shelf language models.  Evaluations compare several of these models.

My main difficulty with the paper is that it does not motivate the need to use reinforcement learning in this way.  For example, is this a model of human grammar acquisition?  If so, there should be more discussion of theories of grammar acquisition that might be supported by the these results.

Secondarily, the paper provides extensive explanations of things like self-paced reading and garden path effects that are probably quite familiar to CMCL audiences, and very little explanation of reinforcement learning or what an 'environment' is in this context, which are more unusual.  Also, it is a bit strange to have an example parse with no formal specification of the parsing rules, even if they are taken from another paper.

Finally, there's no discussion of earlier uses of reinforcement learning in incremental parsing, e.g. Le and Fokkens 2017 in EACL.

There might be an idea here, but I think the presentation must be reworked fairly substantially to be acceptable.

More minor: the paper should include citations for claims like 'AC architectures have been argued to be cognitively realistic' on page 8, and other mentions of 'cognitively realistic'.

---

### Official Review · Reviewer_BjvH · 2022-03-26
**Not enough discussion**

**Rating:** 3
**Confidence:** 4

**Review:**

This paper adresses the question of left-corner parsing and the cognitive motivation of reinforcement learning approaches to this problem. The paper first recalls the main aspects of the LC parsing technique and provides  examples. This first part is very pedagogical, but could be reduced and leave place for the core of the paper. The paper then describes the experiment by presenting first what authors call the environment setup. The idea consists in extracting from the PTB several trees and analyze the behavior of a parser. At each step, depending on the action selected by the parser, a penalty is associated with the state. The experiment itself consists in studying the behavior of 15 reinforcement learning algorithms, applied to a subset of trees chosen for their simplified structures (reduced depth, all starting with an NP-subject). A description of the main characteristics of each RL technique is given, and a  summary of the results with a short discussion is proposed.

The paper presents an interesting and important idea, proposing a method and some results in the perspective of arguing in favor of a cognitive ground for RL algorithms. However, it suffers from several drawback. First, the method for selecting the penalty is not well motivated, and looks arbitrary. The same for the proposed evaluation metrics, that are very basic. But the main problem of the paper is that it fails at providing any evidence in favor of a cognitive motivation. There is no precise analysis of the core of  RL mechanisms that should be arguments in favor of that. The paper only gives a set of results, with no precise discussion and no conclusion. At this stage, even though the idea is interesting, it seems to me not elaborated enough to be presented at CMCL.

---

### Decision · Program_Chairs · 2022-03-29

Reject